# Identifying High-Risk Tumors within AJCC Stage IB–III Melanomas Using a Seven-Marker Immunohistochemical Signature

**DOI:** 10.3390/cancers13122902

**Published:** 2021-06-10

**Authors:** Robin Reschke, Philipp Gussek, Mirjana Ziemer

**Affiliations:** Department of Dermatology, Allergology and Venereology, University Medical Center, 04103 Leipzig, Germany; robin.reschke@medizin.uni-leipzig.de (R.R.); philipp.gussek@medizin.uni-leipzig.de (P.G.)

**Keywords:** biomarker, relapse, immunotherapy, targeted therapy, melanoma

## Abstract

**Simple Summary:**

Immunotherapy and targeted therapy are widely accepted for stage III and IV melanoma patients. Clinical investigation of adjuvant therapy in stage II melanoma has already started. Therefore, methods for relapse prediction in lower stage melanoma patients apart from sentinel node biopsies are much needed to guide (neo)adjuvant therapies. Gene scores such as the “DecisionDx-Melanoma” and the “MelaGenix” score can help assist therapy decisions. However, a seven-marker immunohistochemical signature could add valuable feasibility to the biomarker toolbox.

**Abstract:**

Background: We aim to validate a seven-marker immunohistochemical signature, consisting of Bax, Bcl-X, PTEN, COX-2, (loss of) ß-Catenin, (loss of) MTAP and (presence of) CD20, in an independent patient cohort and test clinical feasibility. Methods: We performed staining of the mentioned antibodies in tissue of 88 primary melanomas and calculated a risk score for each patient. Data were correlated with clinical parameters and outcome (recurrence-free, distant metastasis-free and melanoma-specific survival). Results: The seven-marker signature was able to identify high-risk patients within stages IB-III melanoma patients that have a significantly higher risk of disease recurrence, metastasis, and death. In particular, the high sensitivity of relapse prediction (>94%) in sentinel negative patients (stages IB–IIC) was striking (negative predictive value of 100% for melanoma-specific survival and distant metastasis-free survival, and 97.5% for relapse-free survival). For stage III patients (positive nodal status), the negative predictive value was 100% with the seven-marker signature. Conclusions: The seven-marker signature can help to further select high-risk patients in stages IIB-C but also in earlier stages IB–IIA and be a useful tool for therapy decisions in the adjuvant and future neo-adjuvant settings. Stage III patients with measurable lymph node disease classified as high-risk with the seven-marker signature are potential candidates for neoadjuvant immunotherapy.

## 1. Introduction

The incidence of cutaneous melanoma is rising worldwide [1]. The risk of disease recurrence increases with tumor thickness. For patients with a Breslow thickness of 1 mm (or 0.8 mm and other risk factors such as ulceration, increased mitotic rate or younger age) a sentinel node biopsy for risk stratification and decision of adjuvant therapy should be performed [2,3]. Adjuvant treatment with novel therapies has been shown to significantly prolong the relapse-free survival (RFS) and overall survival (OS) in stage III melanoma [4,5,6,7,8,9]. There are ongoing and planned clinical trials addressing the benefits of novel adjuvant therapy for stage II melanoma patients [10]. The 5-year melanoma-specific survival (MSS) in patients with stages IIB/IIC is, surprisingly, worse than in patients with stages IIIA/IIIB melanoma (87%/82% vs. 93%/83%) [11]. Therefore, it is argued that stages IIB and IIC should also be considered for adjuvant therapy [12]. Staging according to *American Joint Committee on Cancer* 8th edition (AJCCv8) might not be sufficient to classify risk subgroups. For stage I and II melanoma, 5-year MSS rates of 97% and 94% were reported (11). Nonetheless, a significant number of patients die from thin melanomas (<1 mm, IB). This suggests that for a high-risk subgroup in stages I–IIA, adjuvant therapy would also be reasonable [13]. Moreover, a worse outcome for patients with recurrent melanoma after negative sentinel lymph node biopsy was documented compared to sentinel-positive patients [14,15]. Therefore, further prognostic biomarkers must be investigated and validated to reliably identify high-risk patients, especially within lower AJCCv8 melanoma stages such as IB–IIA [16]. We tested and validated the immunohistochemical seven-marker signature consisting of five risk markers (Bax, Bcl-X, COX-2, PTEN, presence of CD20 positive B-lymphocytes) and two protective markers (loss of MTAP and ß-Catenin) for clinical (routine) use in our patient cohort. The signature was found to be an independent negative predictor for OS and RFS and could separate patients into low-risk and high-risk groups [17].

## 2. Materials and Methods

### 2.1. Patient Cohort

Consecutive patients diagnosed with stages IB-III cutaneous melanoma and resected at the university hospital Leipzig between August 2006 and January 2015 were included retrospectively in this study. From a study collective of 299 patients, 88 patients with available formalin-fixed paraffin-embedded (FFPE) primary melanomas were investigated (Figure 1). The clinicopathological characteristics of those patients are given in Table 1.

### 2.2. Immunohistochemical Expression Analysis

In this analysis, whole S-100 positive melanoma FFPE tissue sections were used. The 7-marker signature (immunoprint^®^) determination and scoring was previously described in detail [17]. The analysis was conducted in cooperation with Synvie GmbH, Munich, Germany. The stained slides were evaluated by an external dermatohistopathologist (AG) and two experienced Synvie lab scientists (SS and PN) in a blinded manner. In case of discordant scoring results, a consensus score was assigned. Immunoreactivity was evaluated using an algorithm resulting in a stepwise scoring system (0 to 3+; 0 (negative): 0% positive cells, 3+: greater than 50% positive cells) [17].

### 2.3. Calculation of the 7-Marker-Signature Risk Score

The 7-marker signature risk score was calculated as previously described by a linear combination of the marker coefficient and the corresponding IHC measurements normalized by the number of markers measured [17]. Based on this risk score, patients were assigned to a high-risk group and a low-risk group using the originally described cut-off of 0.135. Risk score values < 0.135 are considered low-risk; values > 0.135 represent high-risk status (Appendix A).

### 2.4. Statistical Analysis

Statistical analyses were conducted by Staburo GmbH, Munich, which matched the Synvie lab risk score results with the survival information provided from the *Clinical Cancer Registry* of Leipzig. Using R statistical software version 4.0.4. *p*-values lower than 0.05 were considered to indicate statistical significance. Appropriate statistical tests (e.g., Fisher’s exact test and *t*-test) were used to analyze differences between high-risk and low-risk study groups. Hazard ratio (HR) with 95% confidence interval (CI) were calculated using Cox regression analysis. For the MSS calculation, only melanoma-related deaths were considered. RFS events included all types of events including local (lymph node, satellite, and in-transit) metastasis, distant metastasis (separately documented for distant metastasis-free survival (DMFS) and MSS events. Kaplan–Meier analyses were performed in order to estimate survival probabilities of the high-and low-risk group. The data-cut-off for survival was June 2019. Patients who died after the 5-year follow-up of unknown causes were censored at date of death for MSS, DMFS and RFS. Subsequently, log-rank tests were used to compare survival rates. In order to analyze further correlations between variables, univariate and multivariable cox regression were used. Univariate analyses were conducted for the following variables: tumor thickness, ulceration, nodal status, gender, and age. In case of significant values in univariate analyses, a multivariable analysis was performed for risk models including previously found risk factors with the strongest impact on melanoma outcome (tumor thickness, nodal status, ulceration) with and without the 7-marker signature.

### 2.5. REMARK

The results of our retrospective study were reported according to the REMARK (Reporting Recommendations for Tumor Marker Prognostic Studies) guidelines [18].

## 3. Results

By using the signature’s cut-off, from the 88 patients, 40 patients were characterized as low-risk and 48 as high-risk (Figure 2, Appendix A).

Almost half of our patients had stages IB–IIA melanomas (*n* = 40). Of these, 8/24 stage IB (33%) and 11/16 stage IIA (69%) were classified as high-risk. In stage IIB, 7/20 melanomas (35%), and in stage IIC, 7/9 (78%) melanomas qualified as high- risk. The only stage IIIA melanoma was stratified as low-risk. Four of five stage IIIB melanomas presented as high-risk (80%). Eleven of 13 melanomas of stage IIIC patients were labeled high-risk (85%). Twenty-one of 47 melanomas without ulceration (45%) qualified as high-risk, whereas 27 of 41 melanomas with ulceration (66%) were stratified as high-risk (*p*-value 0.0557). An equal distribution of low- and high-risk (*n* = 36 vs. *n* = 33) was found amongst patients with stages IB–IIC negative nodal status. However, of patients with positive nodal status (*n* = 19), almost four times more melanomas were classified as high-risk (*n* = 15 vs. *n* = 4 low-risk) (Appendix A). Results showed that higher values of the seven-marker signature score were significantly associated with shorter MSS in a univariate Cox analysis (HR = 2.049; 95% CI = 1.212 to 3.463; *p* = 0.0074). Furthermore, tumor thickness and positive nodal status were significant risk factors of MSS (*p* = 0.0011 and *p* = 0.0027), but ulceration was not. In a multivariable Cox regression model including these prognostic factors/variables (and ulceration) for MSS (Table 2), only tumor thickness remained significant with a *p* < 0.05 (HR = 1.208; 95% CI 1.045 to 1.398 *p* = 0.0108). The *p*-value for the seven-marker signature resulted in *p* = 0.0600 and failed significance by a narrow margin. In a multivariable analysis for DMFS, the seven-marker signature remained significant beside tumor thickness (HR = 1.808; 95% CI = 1.159 to 2.819; *p* < 0.0090) (Appendix A). Both the numerical and the dichotomous risk score was utilized for Cox regression analyses with the endpoint RFS. Univariate Cox regression showed that seven-marker signature, tumor thickness and nodal status were associated with recurrence (all HR ≥ 1.25; all *p* < 0.0005). A multivariable Cox regression using the dichotomous risk score (Table 2) demonstrated that besides nodal status and tumor thickness, the seven-marker signature was a significant independent risk factor for relapse (HR = 30.604 95% CI 4.112 to 228.757; *p* = 0.0008). Receiver operating characteristic (ROC) curves for the risk score and the reference model could be calculated for the endpoints MSS and RFS (Figure 3).

The model for MSS extended by the seven-marker signature proved to be superior compared to the reference model in terms of area under the curve (AUC) (70.9% vs. 62.8%) (Figure 3A). ROC curves with the endpoint RFS (using the dichotomous risk score) also showed higher values for the seven-marker signature model in AUC (76.4% vs. 64.7%) (Figure 3B). For survival estimation/comparison of patients with low- and high-risk seven-marker signature score, Kaplan–Meier (KM) curves were calculated for the endpoints MSS, DMFS, and RFS. As depicted in the respective KM-plots (Figure 4), a high-risk seven-marker signature score led to a significantly lower survival probability in all endpoints—MSS, DMFS, and RFS (at five years, 79%, 57% and 48% for high risk, and at ten years, 61%, 42%, and 28% vs. 100% for low-risk; *p* < 0.0001). In the low-risk group (40 patients), only one locoregional relapse occurred. In sum, in the high-risk group all events but one were predicted correctly by the seven-marker signature. In this cohort, the seven-marker signature had a sensitivity and negative predictive value (NPV) of 100% for MSS and DMFS, and 97% and 97.5%, for RFS, respectively. Of 48 patients with a high-risk seven-marker signature score, 16 had an MSS, 25 a DMFS, and 32 an RFS event. This results in a specificity and a positive predictive value (PPV) for MSS of 55.6% and 33.3%, for DMFS of 63.5% and 52.1%, and for RFS of 70.9% and 66.7%. Thus, approximately two-thirds of the patients with a high-risk seven-marker signature, but only one patient with a low-risk seven-marker signature had an event. To evaluate patients with low(er) risk for metastasis and disease recurrence according to AJCCv8 staging, we performed separate KM analyses of 40 IB–IIA patients (Figure 5, Appendix A). Approximately half of the patients (*n* = 19) showed a high-risk result. For all endpoints, MSS, DMFS and RFS, there was a statistically significant difference in survival of patients with low- and high-risk signature (*p* = 0.029; *p* = 0.0023; *p* = 0.00035). Of the 19 high-risk patients, four had an MSS, seven a DMFS and nine patients an RFS event. All events were correctly indicated by a high-risk score of the seven-marker signature at the same time that no patient with a low-risk score had an event (sensitivity and NPV = 100%) (Table 3). To evaluate the benefit of the seven-marker signature for patients stratified by nodal status, we conducted KM analyses by patient subgroups with negative and positive nodal status. The KM analysis of the nodal status negative patients (Figure 6, Appendix A) yielded significant differences for survival in the high- vs. the low-risk group for the endpoints MSS, DMFS and RFS (*p* = 0.0016; *p* < 0.0001; *p* < 0.0001). Of 69 patients with negative nodal status, 33 were stratified as high-risk and 36 as low-risk according to the seven-marker signature. Among the high-risk patients, MSS events occurred in eight patients, DMFS events in 14 patients and RFS events in 18 patients. In the low-risk group, one patient had an RFS event (sensitivity/NPV: 94.7%/97.2%), but none had a DMFS or MSS event (sensitivity/NPV: 100%/100%). The highest specificity (70%) of the seven-marker signature was reached for the endpoint RFS among patients with negative nodal status. In contrast, KM-plots for patients with positive nodal status (Figure 7, Appendix A) showed significantly different survival probabilities for DMFS and RFS (*p* = 0.02; *p* = 0.01), but not for MSS (*p* = 0.11). Again, the seven-marker signature could predict all events (MSS, DMFS and RFS). Of 15 high-risk patients with positive nodal status (stage IIIA-IIIC), eight had an MSS, eleven a DMFS and thirteen an RFS event. In the four low-risk patients, no melanoma-specific event occurred. Considering all KM analyses of all subgroups with the endpoint RFS among patients with positive nodal status, the highest test quality values were achieved with 66.7% and 86.7% for specificity and PPV, and 100% for both sensitivity and NPV.

## 4. Discussion

The seven-marker signature was initially developed and validated in German melanoma cohorts of Regensburg and Hamburg to be an independent prognostic tool for shorter OS and RFS [17]. This study aimed to investigate (and validate) the seven-marker signature as a prognostic risk tool in an independent melanoma cohort from Leipzig. The seven-marker signature was examined in 88 stage I–III melanoma patients with complete clinicopathological and a minimum of 5-year follow-up data. Multivariable Cox regression analysis was performed to evaluate whether the seven-marker signature contributes prognostic information independent of tumor thickness, ulceration, lymph node status, age and gender. In the multivariable model, the seven-marker signature remained consistently significant for DMFS and RFS and thus was a statistically independent prognostic factor. The seven-marker signature indicated all MSS and DMFS events and all but one RFS event. The sensitivity for MSS, DMFS and RFS was 100%, 100% and 97.0%, respectively. A low-risk signature result could almost exclude subsequent recurrence and metastasis (NPV of 100% for MSS and DMFS, and 97.5% for RFS). Compared with larger cohorts in the literature, in terms of AJCCv8 stages, the cohort studied here has fewer stage IB patients, a relatively high proportion of stage II patients, and a comparable proportion of stage III patients [19,20,21]. The KM analyses of stages IB–IIA underline the predictive potential of the seven-marker signature for disease outcome. For AJCCv8 stages IB and IIA, high values for 5-year MSS of approximately 88–94% and RFS of 76–90% are reported in the literature [11,12,22]. In our study, the 5-year-survival differed significantly in low-risk and high-risk classified patients. For all endpoints (MSS, DMFS and RFS), 5-year survival was 100% in low-risk patients. MSS, DMFS and RFS were significantly lower in high-risk patients (at 5 years, 84%, 63% and 63%, and at 10 years, 77.2%, 63.2% and 44.2%). Accordingly, our results show even better discrimination of low-risk and high-risk patients compared with the seven-marker signature of the subgroup analysis (patients with <2 mm tumor thickness) in the discovery cohort, in which a 5-year OS and RFS of 91% and 92% for low risk and 79% and 57% for high-risk were found [17]. The majority of patients with thin melanoma (stages IA, IB, and IIA) have a 10-year MSS of 98%, 94%, and 88%, respectively. Despite the relatively lower intrinsic mortality risk, the overall number of deaths from thin melanomas is similar or even higher than from thicker melanomas or melanomas diagnosed at higher stages (>stage IIA) due to their high incidence [13,23]. This highlights the importance of identifying high-risk patients especially in lower stages for follow-up care optimization and reasonable adjuvant therapy [24]. Thus, a study focusing on stage IB melanomas showed that computer-assisted multidimensional primary tumor measurements may be useful in predicting patients at highest risk of recurrence [25]. Another study proved the sentinel tumor burden of >0.5 mm to be an independent risk parameter for MSS [26]. Our results also show that risk assessment with the seven-marker signature could be expanded to patients with negative nodal status, including stage IB–IIC patients. Prospectively, once classified as high-risk by the seven-marker signature, indication for adjuvant therapy could also be discussed for stages I and II. Patients also could benefit from a risk-adapted follow-up regimen and imaging. Together with our results in patients with positive nodal status (stages IIIA–IIIC), this suggests that the seven-marker signature allows classification of patients at “high-risk” for disease recurrence independently of nodal status with a sensitivity of 94.4% to 100%, and therewith, may add valuable diagnostic information in addition to sentinel node biopsy. In patients with low-risk melanomas, disease recurrence can almost be excluded (NPVs of 97.2% (nodal negative) to 100% (nodal positive)). Currently, 9–21% of patients with negative sentinel node will experience disease recurrence [15,27]. In our study, 26% of patients with negative nodal status had a disease recurrence, of which all but one could be identified by a high-risk seven-marker signature. We observed a high sensitivity of relapse prediction (>94%) in stages IB–IIC. The sensitivity of a 31-gene and 8-gene score (such as the “DecisionDx-Melanoma” and “MelaGenix” scores) has been reported to be only 64–79% and 32–76% for stages I–II melanoma [19,28,29,30,31,32]. At the same time, the specificity of the seven-marker signature was 70%, similar to the reported specificity of the 13- and 8-gene scores with 70–86% and 43–77%.

### Strengths and Limitations

Compared to the previously published study, where tissue micro array spots were analyzed, we used entire tissue sections for staining with the biomarkers, which allows for a broader and more representative understanding of the melanoma histology. The seven-marker signature reliably identified a subgroup of patients with an increased risk of relapse with PPVs for RFS ranging between 47.4% (stage IB–IIA) and 64.6% (all stages IB-III). The highest PPV of 86.7% was observed in patients with positive nodal status (Table 3). Lower PPVs were observed for the endpoint MSS (e.g., 33.3% for the whole cohort, 21.1% for stage IB/IIA) and may be partially explained by the short length of the follow-up (minimum 5 years). Patients may still experience late metastasis with subsequent death after the last follow-up of this study. The high NPV (97.5% to 100%) observed in the different subcohorts could support rule-out decisions if therapy and follow-up management is concerned. A limitation of the study is the relative high number of dropouts due to no available tissue.

## 5. Conclusions

The results of this study need to be further validated in larger cohorts, and ideally in prospectively collected cohorts (a prospective study with more samples). A prospective (biomarker) study would avoid a follow-up-bias in favor of patients with metastatic events. A prognostic marker/signature should reliably identify patients at high-risk of recurrence/metastasis and, thus, patients who could benefit from adjuvant therapies or adapted follow-up. Therefore, high sensitivity is of primary importance. In addition, melanoma-specific deaths were predicted consistently as high-risk by the seven-marker signature. Prospectively, upcoming neoadjuvant treatment regimens could be offered more individually to high-risk patients in the future. One could imagine that sentinel lymph nodes labeled with a magnetic seed detector would be removed after neoadjuvant treatment, a concept that has already proven to be successful for patients with measurable index nodes [33]. Thus, the seven-marker signature could serve as a stratification tool to guide decision making, particularly for stage I and II melanoma patients, and help in creating a more personalized medicine as well as avoiding invasiveness of therapy, frequency of follow-up, hospitalization and treatment costs in patients stratified as low-risk [34,35,36,37]. The seven-marker signature proved to be a valid prognostic tool to reliably classify patients with increased recurrence and metastatic risk, independent of AJCCv8 staging.

## Figures and Tables

**Figure 1 cancers-13-02902-f001:**
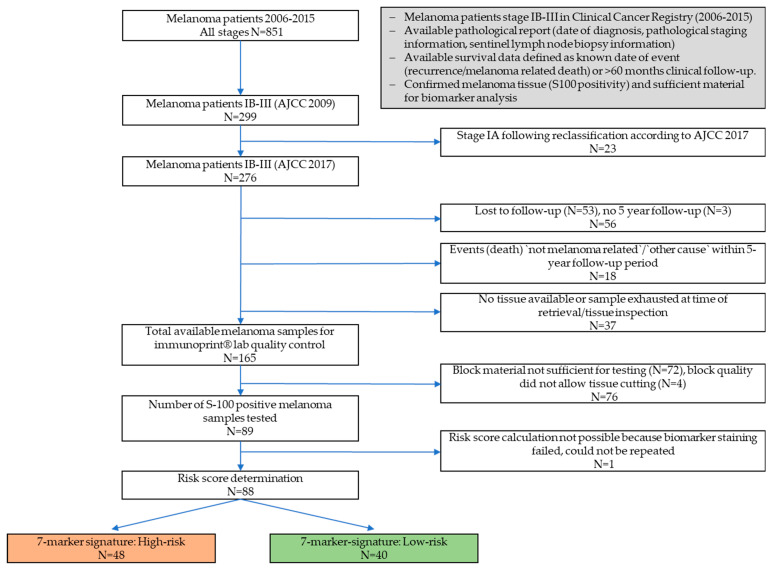
Flow chart depicting the workflow from initial sample collection until generation of final study cohort, staining and statistical evaluation of the 7-marker signature.

**Figure 2 cancers-13-02902-f002:**
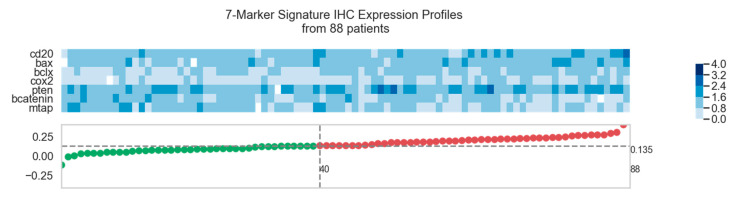
7-marker signature immunohistochemical (IHC) expression profile. The figure shows the IHC expression profile of all tumor specimens from the cohort ordered by their predicted risk score. Each column represents an individual patient consisting of the expression values of the 7-marker signature (5 risk markers and 2 protective markers). The corresponding risk score is plotted for the 40 low-risk patients (green) and the 48 high-risk patients (red). The horizontal line (at 0.135) corresponds to the signature’s cut-off value that separates the groups. The IHC expression values are scaled between 0 (light blue) and 3 (dark blue) according to the used stepwise scoring system (the expression value of 4—as originally described—did not occur in this particular cohort).

**Figure 3 cancers-13-02902-f003:**
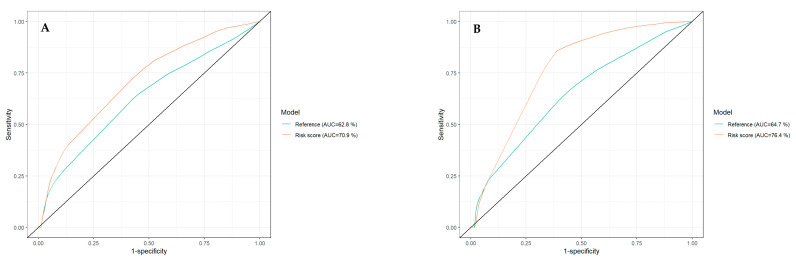
ROC (receiver operating characteristics) curves for the two prognostic models (reference vs. seven-marker signature model) with the endpoints MSS (**A**) and RFS (**B**). The plots illustrate varying prognostic abilities, i.e., sensitivity and specificity, for the seven-marker signature and conventional prognostic factors, such as tumor thickness, ulceration, and nodal status. As a measure of quality comparing the different models, area under the curve (AUC) can be calculated. A (endpoint MSS): with an AUC of 70.9% the model including the seven-marker signature shows superiority over the reference model (AUC = 62.8%). B (endpoint RFS): the model including the seven-marker signature showed a higher AUC (76.4% vs. 64.7%).

**Figure 4 cancers-13-02902-f004:**
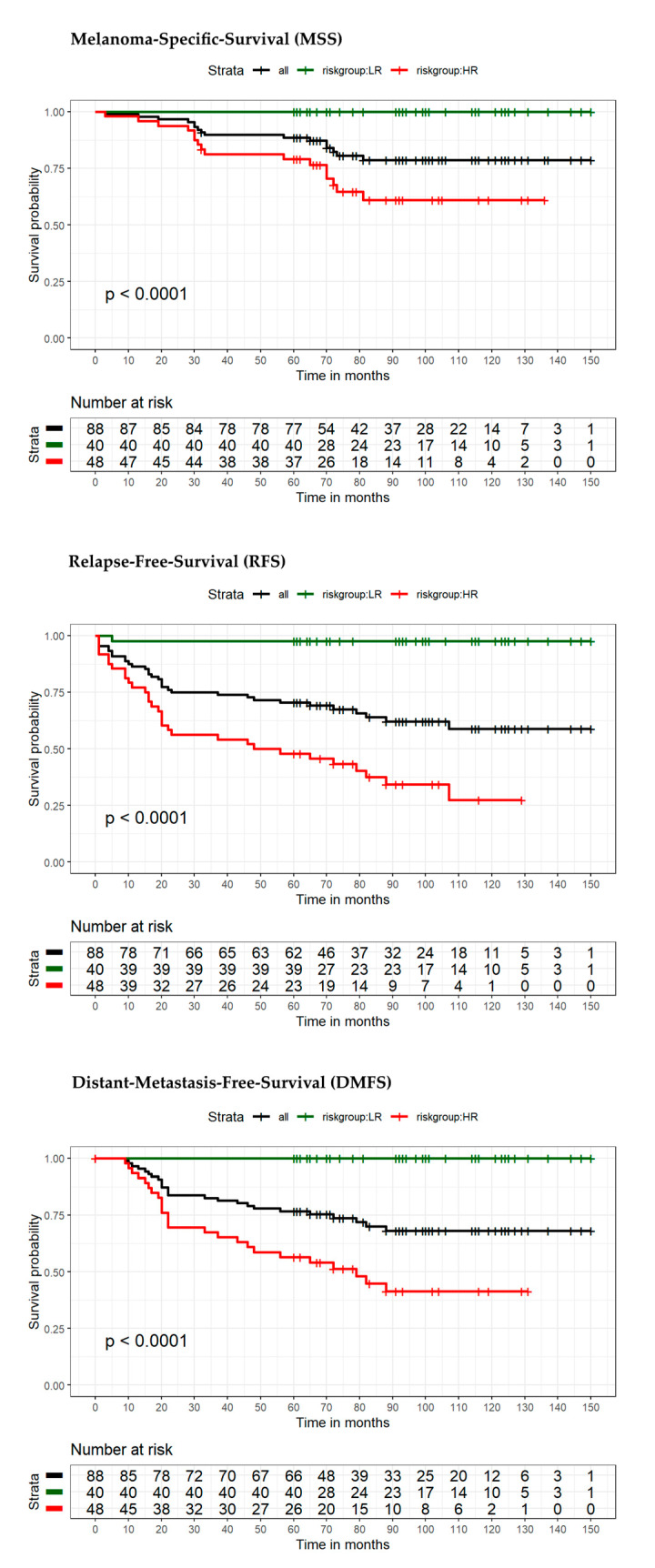
Kaplan–Meier analysis of the entire cohort (*n* = 88 patients; AJCCv8 stages IB—IIIC) stratified with the seven-marker signature into-high risk or low-risk. Estimates of event-free survival for the endpoints MSS, RFS and DMFS at 5- and 10-year follow-up are displayed for: high-risk (in red), low-risk (in green), and all patients (black). Corresponding tables with the number of patients at risk are given below.

**Figure 5 cancers-13-02902-f005:**
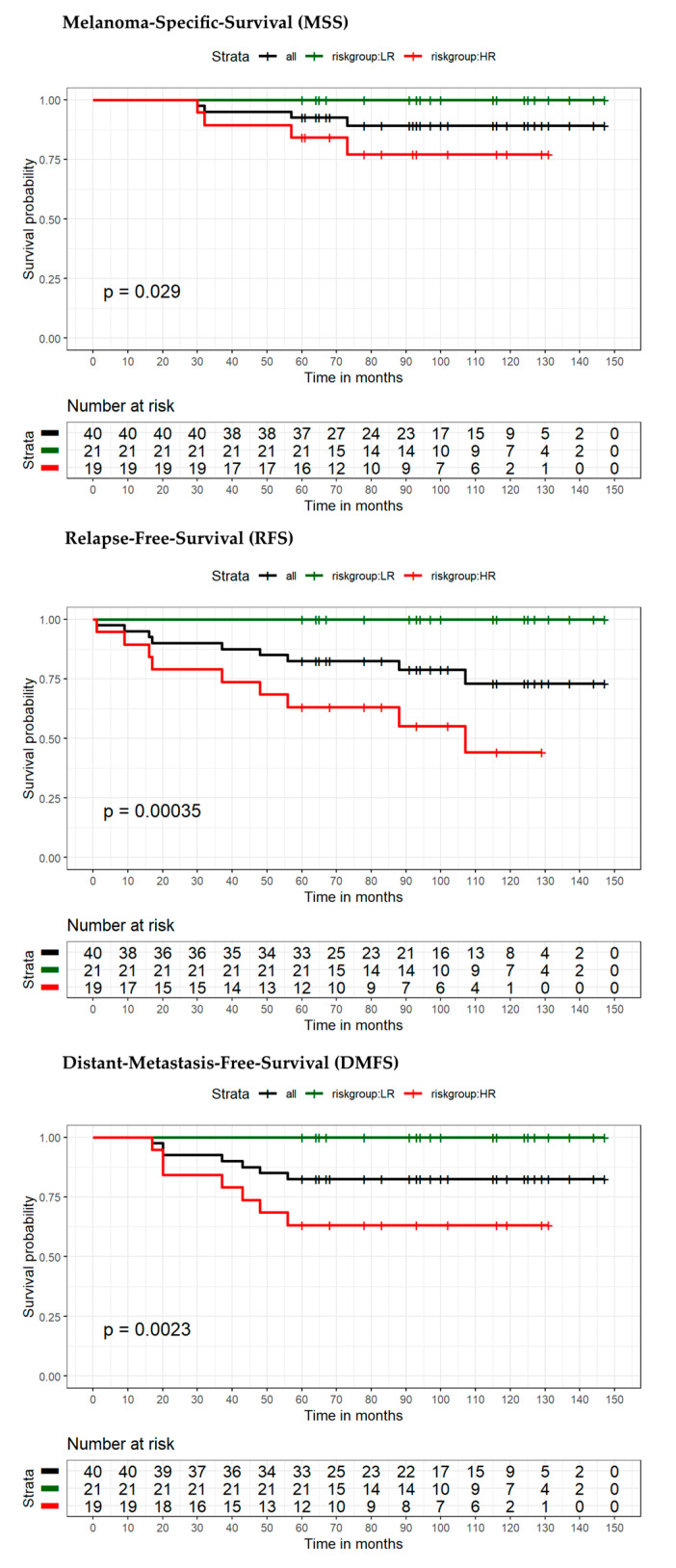
Kaplan–Meier analysis of the sub-cohort with AJCCv8 stages IB—IIA stratified into high-risk or low risk-by the seven-marker signature. Survival estimates for the endpoints MSS, RFS, and DMFS at 5- and 10-year follow-up are displayed for: high-risk (in red), low-risk (in green), and all patients (black).

**Figure 6 cancers-13-02902-f006:**
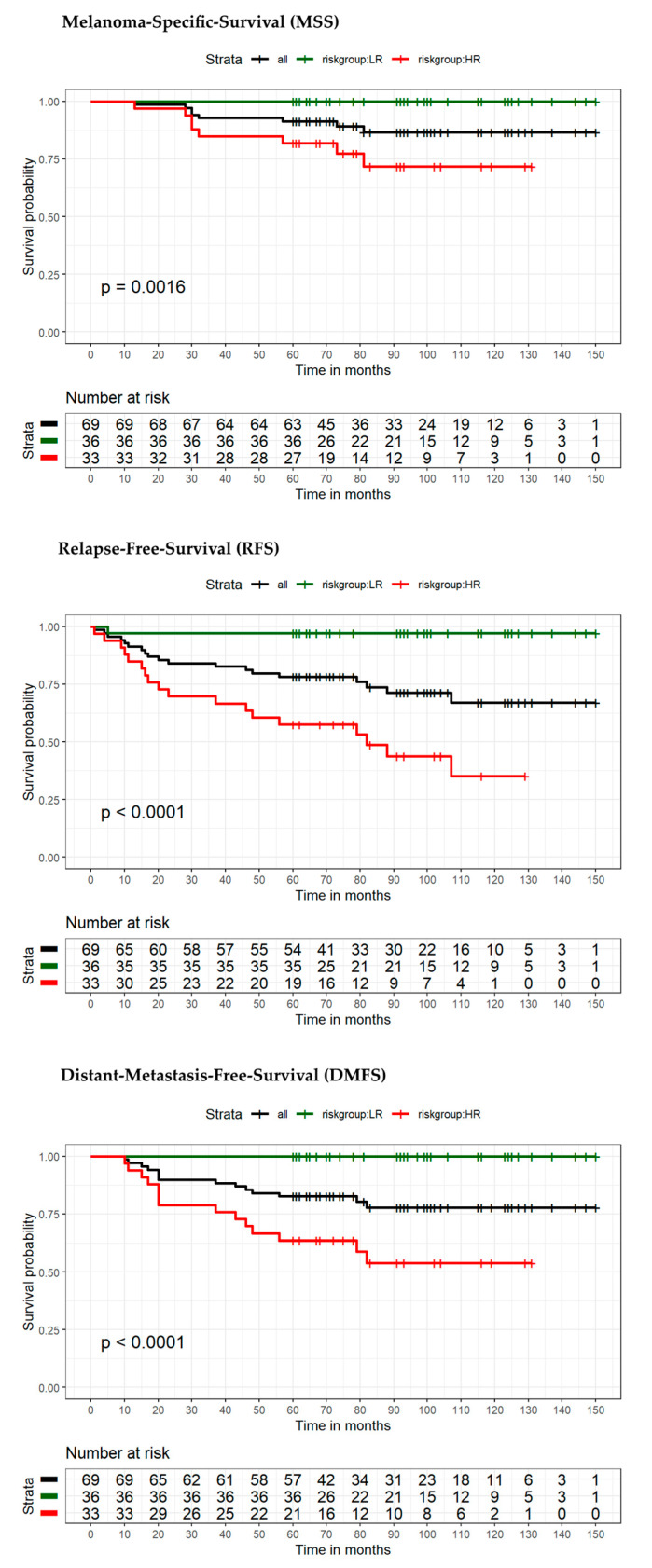
Kaplan–Meier survival estimates of patients with negative nodal status stratified into high-risk and low-risk by the seven-marker signature. Endpoints were MSS, RFS, and DMFS at 5- and 10-year follow-up. High-risk (in red), low-risk (in green), and all patients (black). Corresponding tables with the number of patients at risk are given below.

**Figure 7 cancers-13-02902-f007:**
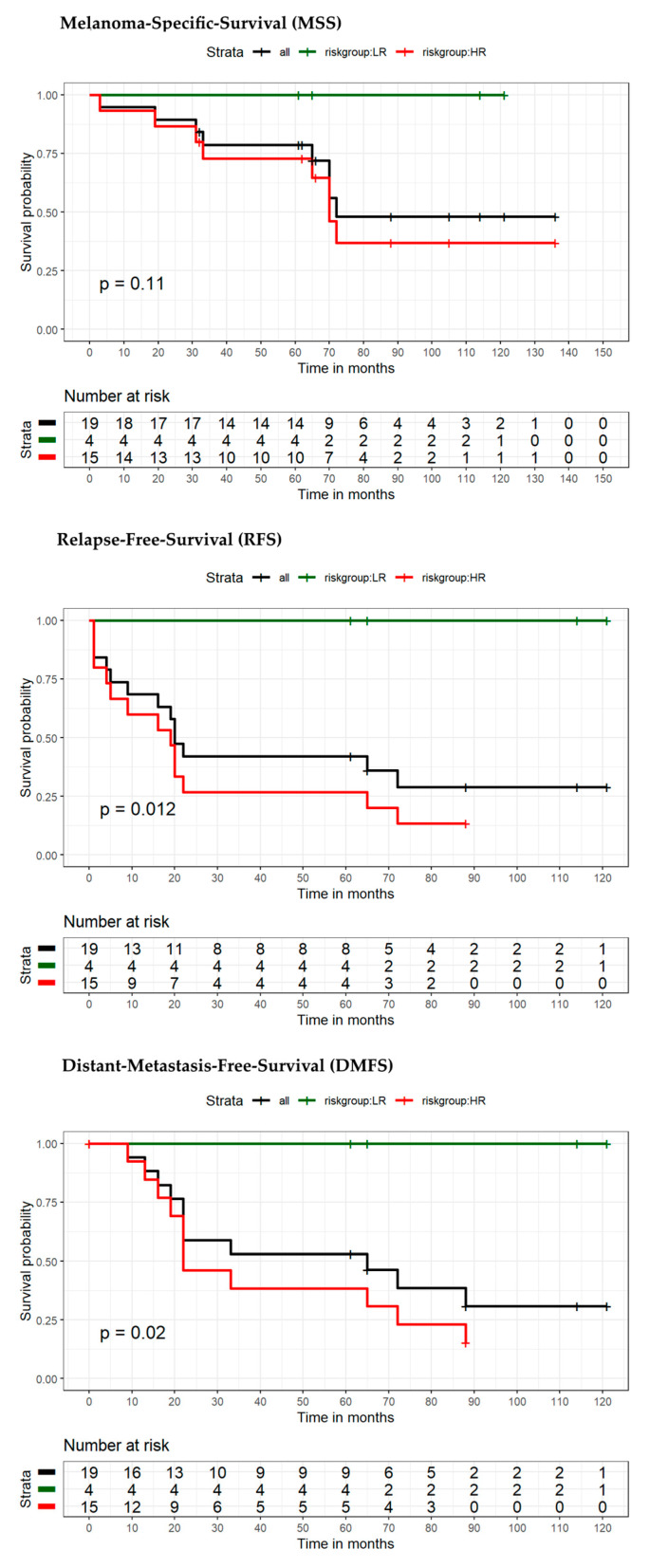
Kaplan–Meier survival estimates of patients with positive nodal status stratified into high-risk and low-risk by seven-marker signature. Endpoints were MSS, RFS, DMFS at 5- and 10-year follow-up. High-risk (in red), low-risk (in green) according to the seven-marker signature, and all patients (black). Corresponding tables with the number of patients at risk are given below.

**Table 1 cancers-13-02902-t001:** Patient and clinicopathological characteristics of the low-risk and high-risk group (all, stage IB–IIIC).

Variable	Category	Overall Patients (*n* = 88)	Low-Risk Patients (*n* = 40)	High-Risk Patients (*n* = 48)	*p*-Value
**Follow-up time (months)**	Mean (SD)		94.9 (28.5)	73.1 (33)	**0.0013**
	Median (min, max)		93.5 (60–150)	71 (3–136)	
**Gender**					0.8304
	Male	48 (54.6%)	21 (23.9%)	27 (30.7%)	
	Female	40 (45.5%)	19 (21.6%)	21 (23.9%)	
**Age (years)**	Mean (SD)		63.4 (17.6)	64.1 (16.2)	0.8556
	Median (min, max)		69.5 (17–90)	66 (27–93)	
**Tumor thickness**	Mean (SD)		2.6 (1.7)	3.9 (3)	**0.012**
	Median (min, max)		2.1 (1.1–8)	3 (1–14)	
**Ulceration**					0.0557
	Yes	41 (46.6%)	14 (15.9%)	27 (30.7%)	
	No	47 (53.4%)	26 (29.5%)	21 (23.9%)	
**SLN status**					n.c.
	Positive 1	19 (21.6%)	4 (4.5%)	15 (17%)	
	Negative 0	69 (78.4%)	36 (40.9%)	33 (37.5%)	
	-cN0	6 (6.8%)	3 (3.4%)	3 (3.4%)	
	-pN0	63 (71.6%)	33 (37.5%)	30 (34.1%)	
**pTNM (AJCC8)**					n.c.
	IB	24 (27.3%)	16 (18.2%)	8 (9.1%)	
	IIA	16 (18.2%)	5 (5.7%)	11 (12.5%)	
	IIB	20 (22.7%)	13 (14.8%)	7 (8%)	
	IIC	9 (10.2%)	2 (2.3%)	7 (8%)	
	IIIA	1 (1.1%)	1 (1.1%)	0 (0%)	
	IIIB	5 (5.7%)	1 (1.1%)	4 (4.5%)	
	IIIC	13 (14.8)	2 (2.3%)	11 (12.5%)	

*p*-value: Fisher‘s exact test for categorical variables; *t*-test for continuous variables. In case of less than 5 patients in the subgroups of the risk group by categorical variable, Fisher‘s exact test could not be calculated (n.c.). Significant *p*-values are bold.

**Table 2 cancers-13-02902-t002:** Univariate and multivariable Cox regression analysis.

		Univariate		Multivariable	
Patients/Cohort	Variables/ Prognostic Factors	HR (95% CI)	*p*-Value	HR (95% CI)	*p*-Value
**Entire cohort** (*n* = 88) (Events: MSS: 16 RFS: 32)	**MSS**				
	7-mrs *	2.05 (1.21–3.46)	**0.007**	1.73 (0.98–3.05)	0.060
	Tumor thickness (mm)	1.27 (1.10–1.46)	**0.001**	1.21 (1.05–1.40)	**0.011**
	Ulceration	2.12 (0.77–5.84)	0.146	1.09 (0.37–3.26)	0.874
	Nodal status	4.51 (1.68–12.07)	**0.003**	2.66 (0.91–7.77)	0.074
	Age at diagnosis	1.00 (0.97–1.03)	0.880	---	---
	Gender	1.01 (0.38–2.71)	0.988	---	---
	**RFS**				
	7-mrs	38.68 (5.27–284.15)	**<0.001**	30.60 (4.11–227.76)	**<0.001**
	Tumor thickness (mm)	1.25 (1.13–1.38)	**<0.001**	1.14 (1.02–1.27)	**0.022**
	Ulceration	2.03 (1.00–4.12)	0.051	1.11 (0.51–2.42)	0.792
	Nodal status	3.79 (1.86–7.71)	**<0.001**	2.15 (1.01–4.56)	**0.046**
	Age at diagnosis	1.00 (0.98–1.03)	0.737	---	---
	Gender	0.60 (0.29–1.24)	0.167	---	---

HR—Hazard Ratio, significant *p*-values are bold; 7 mrs— seven-marker signature risk score: high-risk or low-risk; nodal status—N ≥ 1a; gender—male or female. * Due to the absence of MSS events in the low-risk group, the numerical risk score variable (continuous variable) was used for Cox regression analyses (instead of the dichotomous risk score variable). Since the small scale of the metric risk score leads to very high Hazard ratio values, the score was transformed linearly (multiplied by 10). The interpretation of HR for risk score now does not refer to the increase in the score by 1, but by 0.1. For more detailed results of Cox regression analysis, see also Appendix A.

**Table 3 cancers-13-02902-t003:** The event table shows numbers of patients at risk and the distribution of events in the low-risk and high-risk groups with the resulting predictive power of stratification by the seven-marker signature, indicated by sensitivity, specificity, PPV and NPV.

		MSS			DMFS			RFS		
Patients/Cohort		High Risk	Low Risk	Total	High Risk	Low Risk	Total	High Risk	Low Risk	Total
**Entire cohort** (*n* = 88)	No event	32	40	72	23	40	63	16	39	55
	Event	16	0	16	25	0	25	32	1	33
	Total	48	40	88	48	40	88	48	40	88
	Sensitivity	100%	PPV	33.3%	100%	PPV	52.1%	97%	PPV	66.7%
	Specificity	55.6%	NPV	100%	63.5%	NPV	100%	70.9%	NPV	97.5%
	Increase in risk			183.3%			183.3%			177.8%
**Stage IB–IIC** (*n* = 69)	No event	25	36	61	19	36	55	15	35	50
	Event	8	0	8	14	0	14	18	1	19
	Total	33	36	69	33	36	69	33	36	69
	Sensitivity	100%	PPV	24.2%	100%	PPV	42.4%	94.7%	PPV	54.6%
	Specificity	59%	NPV	100%	65.5%	NPV	100%	70%	NPV	97.2%
	Increase in risk			209.1%			209.1%			198.1%
**Stage IB–IIA** (*n* = 40)	No event	15	21	36	12	21	33	10	21	31
	Event	4	0	4	7	0	7	9	0	9
	Total	19	21	40	19	21	40	19	21	40
	Sensitivity	100%	PPV	21.1%	100%	PPV	36.8%	100%	PPV	47.4%
	Specificity	58.3%	NPV	100%	63.6%	NPV	100%	67.7%	NPV	100%
	Increase in risk			210.5%			210.5%			210.5%
**Stage III** (*n* = 19)	No event	7	4	11	4	4	8	2	4	6
	Event	8	0	8	11	0	11	13	0	13
	Total	15	4	19	15	4	19	15	4	19
	Sensitivity	100%	PPV	53.3%	100%	PPV	73.3%	100%	PPV	86.7%
	Specificity	36.4%	NPV	100%	50%	NPV	100%	66.7%	NPV	100%
	Increase in risk			126.7%			130.1%			126.7%

Endpoints: melanoma-specific survival (MSS), distant metastasis-free survival (DMFS), relapse-free survival (RFS); positive predictive value (PPV), negative predictive value (NPV); High-risk, Low-risk.

## Data Availability

The data presented in this study are available within the article.

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
