# Peer review of "Identifying High-Risk Tumors within AJCC Stage IB–III Melanomas Using a Seven-Marker Immunohistochemical Signature"

_cancers, 2021, doi:10.3390/cancers13122902_

Round 1
Reviewer 1 Report
The study is focused on a very important topic, the stratification of melanoma patients utilising biomarkers that can predict risk of recurrence and therefore influence the treatment/follow up of the patients. The relevance of the study is high since this represents a clear unmet clinical needs.
Overall the study is well planned and the conclusions are supported by the data. One interesting aspect that the authors could discuss more is the case of patients with similar risk scores around the cutoff. In fig. 2 would it be possible to highlight the patient with low risk that had an event an the patients with high risk that did not? how was the difference in the 7-markers? Any other element that could explain this?
Minor considerations are typos and some sentences not totally clear (e.g. line 33 "und" and line 48-49 the sentence does not make sense).
Author Response
Please, find our point by point answers to reviewers comments within the cover letter attached.

Reviewer 2 Report
This is an ably conducted study that applies an immunohistochemical 7-marker signature to identify high-risk melanoma patients. The authors used a retrospective cohort of 88 patients, spanning melanoma stages IB-IIIC, for whom formalin-fixed paraffin embedded primary melanoma tissues were available, along with clinicopathological data and disease follow-up information. Despite the relatively small cohort and the follow-up bias associated with a restrospective, rather than prospective, study the results are quite striking in their ability to match the 7-marker signature risk scores with indicators such as melanoma-specific survival, distant metastasis survival or relapse-free survival, independently of clinical indicators like tumor thickness, ulceration or lymph node status, as well as age and gender. Perhaps more importantly, the model displayed good results for early stage melanoma risk assessment, demonstrating a potential to assist therapy decisions in the near future, should further studies confirm its robustness.
I have only a few minor comments/suggestions:
General suggestion: There are way too many abbreviations in the paper and the reader often gets lost in their meaning. The authors might consider spelling out non-essential abbreviations and perhaps provide an abbreviation list, as I believe this would make the reading easier.
Additional remarks:
1 – Lines 8-14 – The aim of the ‘simple summary’ seems to be providing a ‘state-of-the-art’ overview but it is written in a somewhat misleading manner, and the reader is led to think it is a summary of the work reported in the paper. Perhaps the authors could consider rewriting it to provide a better tie-up to the abstract.
2 –Abstract: I assume databases will retrieve the abstract text but not necessarily that of the ‘simple summary’. Therefore, the abstract should stand by itself and specify the 7 immunohistochemical markers, instead of stating (line 15) ‘this 7-marker signature’.
3 – Abstract, lines 27-28: ‘Stage III patients with measurable lymph node disease classified as high-risk with the 7-marker signature are predisposed for neoadjuvant immunotherapy’. I suggest replacing the word ‘predisposed’ with ‘candidates’.
4- Line 33: editorial remark – replace ‘und’ with ‘and’.
5 – Lines 37-43: This section is written in a somewhat confusing manner. The authors state that ‘The 5-year melanoma-specific survival (MSS) in patients with stages IIB/IIC is surprisingly worse than in patients with stages IIIA/IIIB melanoma’ and then ‘The German Central Malignant Melanoma Registry (CMMR) and the European Organisation for Research and Treatment of Cancer (EORTC) reported worse survival for stages IIIA and IIIB.’ These appear to be contradictory statements and the whole section needs to be clarified.
6- Lines 45-47: Although the 5-year MSS is high in stages I and II (97% and 94%), in total more people die from thin (<1mm, IB) than from thick melanomas (>4 mm) due to their high incidence.’ This may be somewhat misleading because the authors are talking about overall numbers and not the intrinsic mortality risk. Perhaps it could be clarified better. The same applies to lines 275-277.
7 - Line 48: ‘However, number needed to treat would be high’. Something seems to be missing from this sentence. I assume the authors mean ‘…the number of patients considered for treating would be high…’ or something similar.
8 - Line 207: ‘…but no had a DMFS or MSS event…’ I assume the authors mean ‘…none had a DMFS or MSS event…’.
9 – Lines 325-327: ‘…Imaginable, that magnetic seed detector labeled sentinel lymph nodes would be removed after neoadjuvant treatment, a concept already successfully proved for patients with measurable index nodes.’ Something does not seem quite right in this sentence. Please clarify.
Author Response
Please, see the attachment.
